# An Automated Method of 3D Facial Soft Tissue Landmark Prediction Based on Object Detection and Deep Learning

**DOI:** 10.3390/diagnostics13111853

**Published:** 2023-05-25

**Authors:** Yuchen Zhang, Yifei Xu, Jiamin Zhao, Tianjing Du, Dongning Li, Xinyan Zhao, Jinxiu Wang, Chen Li, Junbo Tu, Kun Qi

**Affiliations:** 1Key Laboratory of Shaanxi Province for Craniofacial Precision Medicine Research, College of Stomatology, Xi’an Jiaotong University, 98 XiWu Road, Xi’an 710004, China; yvchen.zhang@outlook.com (Y.Z.);; 2Shaanxi Provincial Key Laboratory of Big Data Knowledge Engineering, School of Computer Science and Technology, Xi’an Jiaotong University, Xi’an 710049, China; cli@xjtu.edu.cn; 3Department of Oral Anatomy and Physiology and TMD, School of Stomatology, The Fourth Military Medical University, Xi’an 710004, China; stevenqk@sina.com

**Keywords:** facial soft tissue landmark, deep learning, object detection, 3D face model

## Abstract

Background: Three-dimensional facial soft tissue landmark prediction is an important tool in dentistry, for which several methods have been developed in recent years, including a deep learning algorithm which relies on converting 3D models into 2D maps, which results in the loss of information and precision. Methods: This study proposes a neural network architecture capable of directly predicting landmarks from a 3D facial soft tissue model. Firstly, the range of each organ is obtained by an object detection network. Secondly, the prediction networks obtain landmarks from the 3D models of different organs. Results: The mean error of this method in local experiments is 2.62±2.39, which is lower than that in other machine learning algorithms or geometric information algorithms. Additionally, over 72% of the mean error of test data falls within ±2.5 mm, and 100% falls within 3 mm. Moreover, this method can predict 32 landmarks, which is higher than any other machine learning-based algorithm. Conclusions: According to the results, the proposed method can precisely predict a large number of 3D facial soft tissue landmarks, which gives the feasibility of directly using 3D models for prediction.

## 1. Introduction

The significance of the aesthetic facial soft tissue in orthodontic treatment has spurred a swift advancement and progression of techniques aimed at quantifying the shape of human facial soft tissue. The identification of landmarks within the facial soft tissue is critical for the precise measurement, assessment, and analysis of the anatomical and morphological characteristics of the human face. Additionally, these landmarks serve as crucial reference points and the foundation for the diagnosis, treatment, and evaluation of clinical work [1,2,3]. Facial landmarks serve a crucial function in facilitating tooth alignment, establishing the occlusal vertical distance, determining the 3D median sagittal plane, analyzing maxillofacial asymmetry [4], aiding in preoperative analysis, surgical design, postoperative prediction, and the efficacy evaluation of orthognathic surgery [5,6].

The orthodontic industry has witnessed significant advancements in technology, leading to the widespread adoption of 2D digital scanning as the primary method. Additionally, 3D facial soft tissue scanning technologies such as laser scanning, computerized tomography, and stereophotogrammetry have emerged, allowing for the acquisition of intricate details pertaining to various parameters of human facial soft tissue [7].

The advent of 3D scanning technology forms the foundation for the prediction of facial soft tissue landmarks. As opposed to radiographic techniques utilized to identify facial soft tissue landmarks, the 3D optical scanning of the face circumvents issues such as overlapping anatomical structures and image distortion [8]. The 3DMD system employs hybrid stereophotogrammetry technology to capture three-dimensional surfaces through the stereo-imaging of the patient. The accuracy of this system has been rigorously tested and found to be satisfactory for clinical applications [9,10]. The work of Littlefield et al. and Ma et al. proved that the error of 3DMD technology over space and the time span is almost negligible, so the patient’s facial model obtained by 3DMD can be equivalent to the real model [11,12].

The traditional marking method is manual annotation method, and the limitations of the manual annotation are:High training and time cost for operators familiar with 3D software;High time cost for annotation of large amount of data;Poor consistency and repeatability of landmarks’ determination among different operators.

The computer-implemented automated algorithm for a 3D face landmarks prediction can effectively improve the stability of the results, reduce the dependence on human experience, and increase the accuracy of the markers [13].

There exist three commonly utilized automatic methods for the prediction of 3D facial soft tissue landmarks:Geometric information algorithm: it calculates the location of landmarks by mathematical methods based on geometric features, which is primarily used to determine landmarks with significant geometric features on the human face and can accurately locate the position of a small number of landmarks with significant features [14].Model matching algorithm: it calculates the location of landmarks by constructing candidate combinations of landmarks and utilizing topological relationships [15].Deep learning algorithm: it predicts the location of landmarks by building a deep neural network. Next, we will introduce the recent related work using deep learning methods in detail.

In recent years, deep learning has emerged as a rapidly developing and innovative branch of automatic facial landmark prediction methods. The research on deep learning in 2D image feature recognition has reached a more mature stage [16]. Therefore, algorithms have been developed to convert 3D facial data into various types of 2D images, such as grayscale maps, RGB maps, geometric maps, curvature maps, etc. [17]. These algorithms then apply existing, more mature 2D facial image feature recognition algorithms to determine landmark information before mapping the 2D features back to 3D to obtain the 3D landmark information.

Wang et al. [18] proposed a deep learning algorithm based on the deep fusion features of 3D geometric data, which converts 3D face data into five 2D attribute maps (including a range map, three surface normal maps, and a curvature map), extracts the global and local features of the data by VGG-16, and uses a coarse-to-fine algorithmic strategy to achieve the precise localization of landmarks. The algorithm was applied to the Bosphorus 3D face dataset to determine 22 facial landmarks with an error of 3.37 ± 2.72 mm, and to the BU-3DFE 3D face dataset to determine 14 facial landmarks with an error of approximately 3.96 ± 2.55 mm [19]. However, the method of prediction by converting 3D models into 2D attribute maps not only leads to the loss of original information, but also makes the models more sensitive to subtle changes in environmental factors [20].

In order to further improve the quantity and accuracy of a 3D human facial soft tissue landmarks prediction, we proposed a prediction algorithm directly based on a 3D model of the human face, which can avoid various problems that occur when converting 3D models into 2D attribute maps and increase the number of predictable landmarks as well as their accuracy. The main contributions of this work are the following:We propose a deep learning architecture for predicting the facial soft tissue landmarks based on 3D face models instead of transforming 3D models into 2D images;We propose a prediction method for facial soft tissues landmarks based on 3D object detection, which is able to significantly increase the number of predicted landmarks to 32;Tested on real diagnostic data from hospitals, it achieves more landmarks of prediction and higher prediction accuracy than previous methods.

## 2. Materials and Methods

### 2.1. Datasets

Datasets are critical in the field of the human facial soft tissue landmark prediction, but the current database of the human facial soft tissue scans is extremely limited, with only 100+ patients’ facial scans and 22 landmarks available [21,22]. Therefore, to train and evaluate our method for predicting landmarks, we created a database of 3D human facial soft tissue scan models from the Hospital of Stomatology of Xi’an Jiaotong University. The database contains 500 patient facial scan models annotated by trained physicians with the coordinates of 32 landmarks. Ten of the landmarks labeled in the dataset are left–right symmetric and 22 are individually present. Theoretically, the prediction errors of the left–right symmetric points should be similar, but since the faces are not perfectly symmetric, they can be treated as different landmarks, which will also be given later for verification. A total of 500 subjects between 14 and 23 years of age (228 males and 272 females) were randomly selected in our study from the Hospital of Stomatology of Xi’an Jiaotong University, Xi’an, Shaanxi, China. The 3dMD (3dMD, Atlanta, GA, USA) of all the samples were taken by experienced doctors for diagnosis and treatment requirements between 2018 and 2022. The subjects were excluded if they had maxillofacial trauma, severe asymmetry, and a history of cleft lip and palate. The dataset was divided into training set: test set: validation set = 7:2:1. Table 1 shows the 32 landmarks and their descriptions, while Figure 1 provides schematic diagrams.

### 2.2. Architecture Overview

Given the 3D human facial soft tissue model data, denoted by *G*:(1)$$G=\left[\begin{array}{ccc} x_{1} & y_{1} & z_{1}\\ x_{2} & y_{2} & z_{2}\\ \vdots & \vdots & \vdots \\ x_{K} & y_{K} & z_{K} \end{array}\right]$$
where *K* is an uncertain parameter which leads to an uncertain input dimension if the point set is used directly as the input data. Meanwhile, if a method such as FCN [23] is used to obtain a fixed-size output as the input of the network by full convolution, it is not possible to convolve the points adjacent to each other on the 3D space due to the order of the point set.

Therefore, in order to obtain a fixed input size, we transform the model into a 3D point cloud while doing data normalization, using a 3D tensor to represent the point cloud as $\hat{G}$:(2)$$\hat{G}=\left\{\begin{array}{c} \left[\begin{array}{cccc} A_{1,1,1} & A_{1,2,1} & \ldots & A_{1,M,1}\\ A_{1,1,2} & A_{1,2,2} & \ldots & A_{1,M,2}\\ \vdots & \vdots & \ddots & \vdots \\ A_{1,1,M} & A_{1,2,M} & \ldots & A_{1,M,M} \end{array}\right],\ldots ,\left[\begin{array}{cccc} A_{M,1,1} & A_{M,2,1} & \ldots & A_{M,M,1}\\ A_{M,1,2} & A_{M,2,2} & \ldots & A_{M,M,2}\\ \vdots & \vdots & \ddots & \vdots \\ A_{M,1,M} & A_{M,2,M} & \ldots & A_{M,M,M} \end{array}\right] \end{array}\right\}$$
where *M* is a hyperparameter, it determines the coordinate granularity of the 3D point cloud model; $A_{i,j,k}=\left\{\begin{array}{c} 0\\ 1 \end{array}\right\}$ is used to indicate whether a point exists at coordinates (i,j,k).

Our goal is to find *N*-specified landmarks, which can be denoted as *S*:(3)$$S=\left[\begin{array}{ccc} x_{1} & y_{1} & z_{1}\\ x_{2} & y_{2} & z_{2}\\ \vdots & \vdots & \vdots \\ x_{N} & y_{N} & z_{N} \end{array}\right]$$

The pipeline of this work is shown as Figure 2. The code of this paper will be released at https://github.com/YuchenZhang-Academic/3D-Facial-Landmark, acessed on 24 April 2023. The specific pipeline is described as follows:Transform the 3D human facial soft tissue model into a 3D point cloud model;Input the point cloud into the object detection network to obtain the boxes of the six organs (eyes, nose, lips, chin, right face, and left face, each box is represented by a six-dimension vector);Extract the corresponding coordinate of each organ and put them into their prediction model to obtain the landmarks which need predicting.

### 2.3. Object Detection Network

The number of human facial soft tissue landmark prediction using attribute extraction, dimensionality reduction transformation, and geometric algorithms is generally limited by the algorithm and model size to approximately 20 points [19,24]. To break through this limitation, we first performed the object detection of organs on the 3D model, and then started with each organ (also known as region of interest) separately for the landmark predictions, and we were able to predict up to 32 landmarks.

The goal of object detection is to obtain six 6-dimensional vectors $v_{k}\left(k=0,1,\ldots ,5\right)$:(4)$$v_{k}=\left[x_{k},y_{k},z_{k},xl_{k},yl_{k},zl_{k}\right]$$
where $x_{k},y_{k},z_{k}$ is the center coordinate of the ROI (organ) box; and $xl_{k},yl_{k},zl_{k}$ is the distance between the face and the center of the ROI box. The output vector of the object detection phase is shown in Figure 3. The task of the object detection network is formally described as follows:(5)$$f\left(\hat{G}\right)=\left[v_{eyes},v_{nose},v_{lips},v_{chin},v_{rightface},v_{leftface}\right]$$
where the input dimension is $M^{3}$ and the output dimension is 6×6.

In order to capture the interconnections between the points at the 3D level to better extract the features of the landmarks, we mainly use 3D convolution for the construction of the model. By capturing the features of the facial attributes within the different organs, the network can finally give information on the locations and sizes of the boxes to which the six organs belong. The network architecture of the object detection phase is shown in Figure 4.

The loss function used in this phase is as follows:(6)$$L\left(f\right)=\frac{\sum_{i=0}^{6}\left[\lambda_{1}\sqrt{{\left(x_{i}-{\hat{x}}_{i}\right)}^{2}+{\left(y_{i}-{\hat{y}}_{i}\right)}^{2}+{\left(z_{i}-{\hat{z}}_{i}\right)}^{2}}+\lambda_{2}\left(\sqrt{xl_{i}^{2}-{\hat{xl}}_{i}^{2}}+\sqrt{yl_{i}^{2}-{\hat{yl}}_{i}^{2}}+\sqrt{zl_{i}^{2}-{\hat{zl}}_{i}^{2}}\right)\right]}{6}$$
where $\lambda_{1},\lambda_{2}\left(\lambda_{1}+\lambda_{2}=1\right)$ are the hyperparameters, and their ratio determines the weights for the center error and the box size error.

### 2.4. Prediction Network

After the calculation of the boxes for each organ in Section 2.3, we can partition a 3D human facial soft tissue model into six parts, and then train different network parameters for each part to achieve higher prediction accuracy.

The goal of coordinate prediction is to obtain the coordinates of landmarks for different organs. The normalized input dimension is the same as the object detection stage, and the output is a vector of dimension 3×N, where *N* is the number of points to be predicted in each organ, which is shown in Table 2.

The network architecture of the coordinate prediction stage is similar to Section 2.3, with the difference that Resnet18 [25] is added between the multiple loop convolution and fully connected layers for the more accurate prediction of landmarks, allowing the network to reach greater depths, which increases the training cost but improves the accuracy. The network architecture diagram is shown in Figure 5.

Two kinds of loss functions can be used in this phase:(7)$$L_{1}\left(f\right)=\frac{\sum_{i=0}^{N}\left[\sqrt{{\left(x_{i}-{\hat{x}}_{i}\right)}^{2}+{\left(y_{i}-{\hat{y}}_{i}\right)}^{2}+{\left(z_{i}-{\hat{z}}_{i}\right)}^{2}}\right]}{N}$$

Equation (7) is an error calculation formula given by the Euclidean norm; this error calculation method tends to minimize the average error at each point.
(8)$$L_{2}\left(f\right)=\max_{i=0}^{N}\left[\sqrt{{\left(x_{i}-{\hat{x}}_{i}\right)}^{2}+{\left(y_{i}-{\hat{y}}_{i}\right)}^{2}+{\left(z_{i}-{\hat{z}}_{i}\right)}^{2}}\right]$$

Equation (8) is an error calculation formula given by an infinity norm; this error calculation method tends to minimize the maximum error.

Since both the maximum and average errors are important in the prediction work of human facial soft tissue landmarks predication, the hyperparameters $\lambda_{1},\lambda_{2}$ were also added to the prediction phase to adjust the weights between them. The final error calculation formula is as follows:(9)$$L\left(f\right)=\lambda_{1}L_{1}\left(f\right)+\lambda_{2}L_{2}\left(f\right),\;\lambda_{1}+\lambda_{2}=1$$

### 2.5. Experiment

We first describe the method of data preprocessing, then give the definition and calculation of the loss between the proposed method and manual labeling; finally the equipment used for the experiments as well as the training time and error will be described.

#### 2.5.1. Data Preprocessing

The number of points in the 3D model of human facial soft tissue is uncertain, which necessitates building a homogeneous mesh to accommodate the points in the point cloud. This paper outlines the pre-processing of the data in the following steps:Cleaning the data by removing any obvious occlusions, such as the physician’s hand fixing the patient’s head, and any obviously incorrect markers;Normalizing all the data to the range [−1, 1], while preserving the scaling multiplier *S* of each data for error calculation.Creating a three-dimensional uniform grid with a side length of *M*, which can accommodate $M^{3}$ points, and the data accuracy is $\frac{2S}{M}$;After adjusting the valid numbers of the data, iteratively setting $A_{i,j,k}=1$ for the locations of the points present in the grid.

By using this method, the uncertain input dimension is transformed into an input dimension fixed at $M^{3}$, which can simplify the network. Additionally, experimental evidence has shown that, within certain limits, changing the number of effective digits of data does not affect the integrity of the 3D model.

#### 2.5.2. Loss Calculation

As mentioned in Section 1, the error between the face model obtained by 3DMD technology and the real model is negligible. Therefore, this article assumes that the landmarks manually labeled by doctors are accurate values, and the error is obtained by calculating the Euclidean distance (unit: mm) between the predicted coordinates and the manually marked coordinates to evaluate the prediction effect of the model. Due to the normalization of the data in the preprocessing stage, the previous scaling operation needs to be taken into account when calculating the error. The unit of the original model is mm, so the error calculation method used for the performance evaluation is:(10)$$L=S\cdot \sqrt{{\left(x-\hat{x}\right)}^{2}+{\left(y-\hat{y}\right)}^{2}+{\left(z-\hat{z}\right)}^{2}}$$
where (x,y,z) is the coordinate obtained by the proposed method, $\left(\hat{x},\hat{y},\hat{z}\right)$ is the manually labeled coordinate (accurate value), and *S* is the scaling multiplier recorded in Section 2.5.1, which has different values in each model.

#### 2.5.3. Experimental Setting

In this paper, only the ResNet18 network of the prediction phase uses pre-training parameters, while the rest of the network is trained using random initialization parameters. The experiments conducted in this paper take the aforementioned *M* as M=200, so the dimension of both parts of the input is 200×200×200, while the output dimension of the object detection network is 6×6=36, and the output dimension of the prediction part changes with the corresponding organ, as shown in Table 2.

We used two GeForce 2080Ti for training, and the training time was approximately 15 h for the object detection phase and 10 h for the prediction phase, but when setting batch_size ≤ 10, parallel training can be performed to reduce the training time.

To properly train these models, for the object detection phase, we train the models for 600 epochs, and actually the models converge at the 400th epoch; for the prediction phase, we train each model for 500 epochs, and actually the models converge at around the 350th generation. To avoid overfitting, we choose models that are a few epochs ahead of convergence.

## 3. Results

We will evaluate the model of our work by comparing the error between our work and manual marking and comparing our work’s performance with other works.

### 3.1. Comparison of Errors between Our Work and Manual Marking

First, we calculated the error between our method and manual labeling, and presented the results in a box-line diagram in Figure 6. The landmarks belonging to the same organ are indicated with the same background color, and the left and right landmarks are identically ordered and colored. The diagram clearly shows that the errors are relatively uniform for landmarks within the same organ, but more variable for those on the left and right sides of the face. The causes of this variation are discussed in Section 4.

Overall, our method achieved a mean error of 2.62±2.39 mm compared to manual labeling, with 72.73% of the landmarks automatically located within a mean error of 2.5 mm and 100% within 3 mm. These results demonstrate the effectiveness of our approach in accurately predicting the coordinates of human facial soft tissue landmarks. The comparison of our results with other works is presented in Table 3.

### 3.2. Comparison of Errors between Our Work and Other Works

To further investigate the performance of 3D human facial soft tissue landmark prediction based on object detection, we compared the landmark errors in our experiment with those of other works. The specific comparison results are presented in Table 3. The calculation method of the error is documented in Section 2.5.2. Our work achieves the highest precision for almost all comparable points, while predicting at least 13 additional landmarks. It should be noted that not all the predicted landmarks by the methods mentioned in the table are shown, such as some landmarks around the brows.

## 4. Discussion

We conducted an experiment and analyzed the results based on the organs, symmetry, and maximum and minimum error values.

Regarding organs, the error distribution of landmarks in the eyes, nose, and lips showed a relatively uniform distribution with an error fluctuation of approximately 1 mm. This could be attributed to the good symmetry of these organs and the similar geometric features of each point. As a result, the model was able to effectively extract their features through 3D convolution. In the training process, the error of each point was uniformly reduced to achieve a lower average error. However, the error distribution of landmarks in the chin and face was not uniform due to the lack of obvious geometric features in most of these landmarks and the significant differences in their geometric features.

Regarding symmetry, the errors of most landmarks, except those around the eyes, were uniform, with an error fluctuation of approximately 0.5 mm. The errors of landmarks around the eyes showed fluctuations of about 1 mm, but the errors of the landmarks near the left and right eyes were relatively uniform. This might be due to the 3D model’s asymmetry resulting from an angular deviation due to it not being squarely posed to the camera, causing the yOz plane to divide the model unevenly.

Regarding the maximum and minimum error values, Gonion showed the highest error value, which could be attributed to the susceptibility of this point’s geometric features to individual differences such as the face shape and facial muscle fullness. In contrast, the two other landmarks in the face showed the minimum error values, despite having similar geometric features to Gonion with large angular variation. The geometric features of Tragus and Zygoin were less susceptible to individual differences, and the error calculation tended to minimize the average error of landmarks in an organ. Hence, both the landmarks with the largest and smallest errors appeared in the face organ.

Based on the results of the comparison, the proposed method exhibits the best mean loss among all compared methods. Compared with other methods, the mean error has been reduced by at least 0.59 mm (18.38%) and at most by 2.43 mm (48.12%). In the comparison of a single point, our method achieves the highest accuracy among all methods with 75% (24/32) of the landmarks and 100% of the landmarks with the highest accuracy among the deep learning-based methods. The notable landmarks in this study include Glabella and Cheilion (Left). The proposed method in this paper has significantly improved the accuracy of these two points by 57.48% (from 6.35 mm to 2.70 mm) and 23.58% (from 3.35 mm to 2.56 mm), respectively. Furthermore, in comparison to deep learning methods, the accuracy improvement on Cheilion (left) ranges from a minimum of 34.02% (from 3.88 mm to 2.56 mm) to a maximum of 64.20% (from 7.15 mm to 2.56 mm). These two representative landmarks visually demonstrate the performance of our method in predicting all landmarks. From the perspective of symmetry, our method achieved the lowest average error at the Endocanthion, Exocanthion, Alare, and Cheilion points. Furthermore, it ensures that the errors of the two symmetrical points are nearly identical, thereby demonstrating the method’s ability to maintain both accuracy and stability in predictions. Notably, some points in the methods based on geometric information have accuracies exceeding our methods, such as subnasale and subspinale, which are closer in distance and do not have distinct geometric features. However, by considering their definitions as the median point and the deepest point, we can greatly improve their prediction accuracy. This demonstrates the importance of the interrelationship between landmarks and the model, therefore extracting the global as well as local attributes of the model is an effective method to improve the prediction accuracy.

We also acknowledge the performance of other methods that effectively predict 3D human facial soft tissue landmarks. Wang et al. (2022) utilized the Heatmap Regression with the Graph Convolutional Network method on the BU-3DFE and FRGCv2 databases to predict eight landmarks, achieving an error of 1.97±1.50 and 2.54±1.64, respectively. This demonstrates the ability to extract interrelationships between landmarks using the graph convolutional neural Network [29]. In 2018, Terada et al. proposed a CNN-based method that experimentally predicted 14 landmarks. Through comparative experiments, they found that the ResNet34+Data Augmentation approach yielded optimal results. Although this method involved transforming the 3D model into a 2D attribute map, the experimental results still provided valuable insights [30].

This work provides an alternative approach for predicting human facial soft tissue landmarks that surpasses the translation of 3D models into 2D images. Our findings indicate that superior results can be achieved by directly predicting on 3D models. Moreover, we demonstrate the effectiveness of coarse-to-fine methods such as object detection. The direct manipulation of the 3D model is also feasible in the dental clinical field, such as for landmark prediction in CBCT models, and coarse-to-fine models can be utilized in similar fields.

However, we identified several areas where similar approaches have the potential for further improvement:Existing algorithms do not fully extract the complex interrelationships between points in the 3D model;The data used in this study are insufficient for clinical practice. The further application of this method on a larger patient population is necessary to ensure reliable results. We plan to integrate the development of a 3D human facial soft tissue model database to expand the patient dataset;It is essential to conduct additional research to validate and establish the proposed method as a reliable tool in clinical practice. This entails conducting more comprehensive studies that evaluate its effectiveness, accuracy, and potential limitations in diverse clinical settings.The proximity of some landmarks is so close that, if the error in prediction is not sufficiently small compared to the distance to its nearest point, the prediction of a point becomes meaningless. However, this method of evaluation is not currently employed in corresponding works;There are numerous clinically significant landmarks present in both human facial soft tissue and CBCT images that existing methods are unable to predict due to limitations in algorithms and the corresponding databases.

These are critical issues that require further consideration in future research.

## 5. Conclusions

In this paper, we propose a novel method for predicting the coordinates of 3D human facial soft tissue landmarks. Our approach first performs object detection on the 3D model and divides it into six parts: eyes, nose, lips, chin, left face, and right face. Then, each model of these six parts is used for landmark prediction. Experimental results on real datasets show that the proposed method has a lower mean error and predicts more landmarks than other methods. It has achieved a high accuracy for most landmarks and has good stability. Additionally, we created and continuously updated a database to address the issue of insufficient data in the current 3D human facial soft tissue database. Overall, our method provides a 3D model-based prediction method for 3D human facial soft tissue landmark prediction, and experimentally demonstrates the feasibility of the method and some advantages over other methods in terms of accuracy, stability, and the number of predictions.

In our future research, we plan to further investigate the two-part neural network by exploring the use of graph neural networks (GNNs) [31] and self-cure networks [32]. Moreover, we intend to increase the scale of the network and implement a Transformer architecture [33]. We will also introduce an evaluation method that uses the ratio of the prediction error of a point to the distance from its nearest point as an indicator to provide confidence in the prediction. Furthermore, we will continue to update the database of 3D human facial soft tissue models. Additionally, we consider extracting the attributes of landmarks with high errors and incorporating them into the model as a potential method to improve the accuracy rate.

## Figures and Tables

**Figure 1 diagnostics-13-01853-f001:**
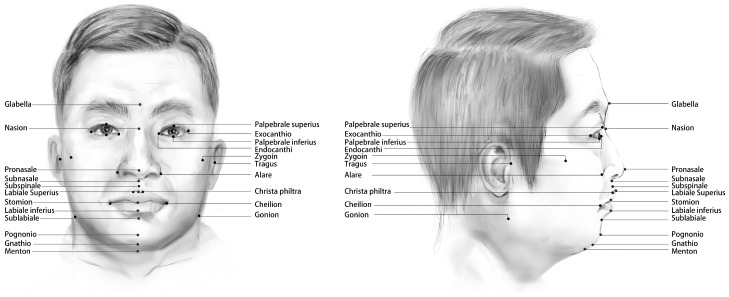
The 32 landmarks predicted in this paper are labeled on the human face.

**Figure 2 diagnostics-13-01853-f002:**
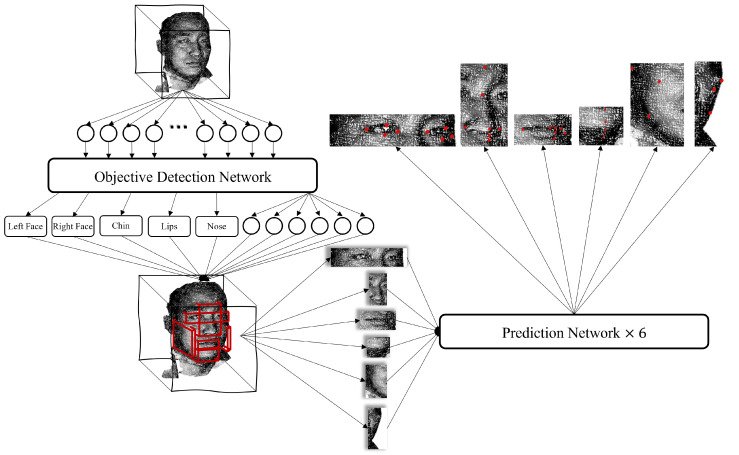
The pipeline of the proposed work. First, the 3D model is transformed into a 3D point cloud, which is partitioned into six parts using an object detection model, where the output vector of each part is [x,y,z,zl,yl,zl]. After that, the points of each part are extracted from the 3D model according to the calculated range of boxes, and the coordinates of landmarks are predicted by the prediction model.

**Figure 3 diagnostics-13-01853-f003:**
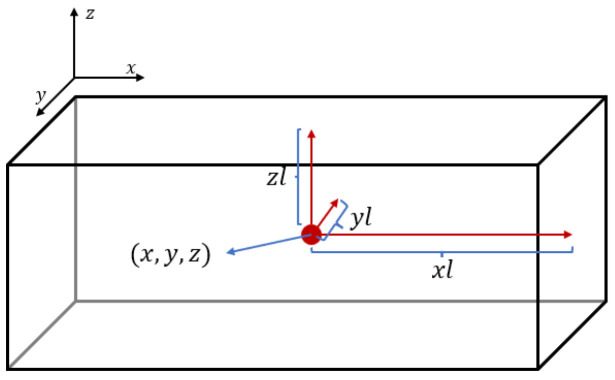
This is a graphical representation of the output vector of the object detection phase. Here, (x,y,z) are the coordinates of the box center and (xl,yl,zl) are the distances from the center to the three faces.

**Figure 4 diagnostics-13-01853-f004:**
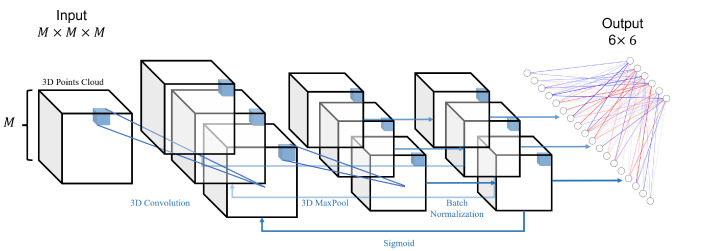
This is the network architecture diagram of the object detection phase. It goes through several cycles in a structure consisting of 3D convolution, 3D maximum pooling, 3D BatchNorm, and Sigmoid, before finally outputting a vector of 6×6 dimensions after a fully connected layer.

**Figure 5 diagnostics-13-01853-f005:**
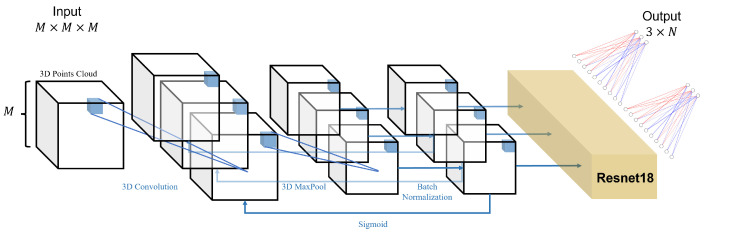
This is the network architecture diagram of the coordinate prediction phase. It goes through several cycles in a structure consisting of 3D convolution, 3D maximum pooling, 3D BatchNorm, and Sigmoid; then, the data will go through the Resnet18 network, and finally output a vector of 3×N dimensions after a fully connected layer.

**Figure 6 diagnostics-13-01853-f006:**
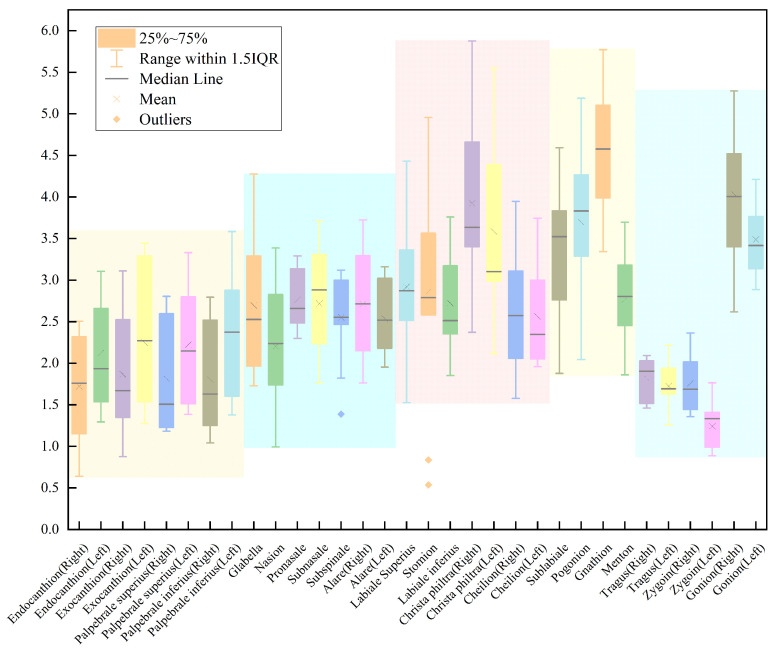
The figure is a box-line diagram of the error between our work and manual labeling, where points that are in the same organ are labeled with the same background color (where the left face and right face are considered as the same organ). The overall error is 2.62±2.39 mm, while 72.73% landmarks are located within a mean loss of 2.5 mm and 100% landmarks are within mean loss 3 mm.

**Table 1 diagnostics-13-01853-t001:** The organs, abbreviations, names, and definitions of the 32 landmarks predicted in this paper.

Organ	Abbreviation	Landmarks	Definition
Eyes	En	Endocanthion (right and left)	The soft tissue point located at the inner commissure of the right eye fissure
	Ex	Exocanthion (right and left)	The soft tissue point located at the outer commissure of the right eye fissure
	Ps	Palpebrale superius (right and left)	Most superior point on the margin of the upper eyelid
	Pi	Palpebrale inferius (right and left)	Most inferior point on the margin of the lower eyelid
Nose	G	Glabella	Most anterior midpoint on the front-to-orbital soft tissue contour.
	Na	Nasion	Point directly anterior to the nasofrontal suture, in the midline
	Pn	Pronasale	The most anteriorly protruded point of the apex nasi
	Sn	Subnasale	Median point at the junction between the lower border of the nasal septum and the philtrum area
	A	Subspinale	The deepest point seen in the profile view below the anterior nasal spine
	Al	Alare (right and left)	The most lateral point on the nasal ala
Lips	Ls	Labiale superius	Midpoint of the vermilion border of the upper lip
	Sto	Stomion	Midline point of the labial fissure when the lips are naturally closed, with teeth shut in the natural position
	Li	Labiale inferius	Midpoint of the vermilion border of the lower lip
	Cph	Christa philtra (right and left)	Point on each elevated margin of the philtrum just before projection to the vermilion line
	Ch	Cheilion (right and left)	Outer corners of the mouth where the outer edges of the upper and lower vermilions meet
Chin	B	Sublabiale	Most posterior midpoint of the philtrum
	Pg	Pogonion	Most anterior median point on the mental eminence of the mandible
	Gn	Gnathion	Median point halfway between pg and me
	Me	Menton	Most inferior median point of the mental symphysis
Face	Tra	Tragus (right and left)	The most convex point of the tragus at the external ear canal
	Zv	Zygion (right and left)	Instrumentally determined as the most lateral point on the zygomatic arch
	Go	Gonion (right and left)	Point on the rounded margin of the angle of the mandible, bisecting two lines—one following the vertical margin of ramus and one following the horizontal margin of corpus of mandible

**Table 2 diagnostics-13-01853-t002:** The output dimensions corresponding to the prediction networks of different organs.

Organ	Output Dimension
Eyes	8
Nose	7
Lips	7
Chin	4
Right face	3
Left face	3

**Table 3 diagnostics-13-01853-t003:** The table shows the error of our work compared to the five remaining methods and compared to manual annotation. All of these are deep learning-based methods except Baksi et al.’s method [24]. Since our work additionally makes predictions for some landmarks, the remaining method gaps are filled with -. The best method for each point is marked in bold.

Landmark	Baksi1 [24]	Fanelli [26]	Zhao [27]	Sun [28]	Wang [19]	Our Method
Endocanthion (right)	3.13±0.84	2.80±2.00	2.90±1.36	3.27±5.51	3.11±2.24	2.12±0.98
Endocanthion (left)	3.80±1.43	2.60±1.80	2.93±1.40	3.35±5.67	2.79±1.63	1.72±1.08
Exocanthion (right)	3.44±1.47	4.00±2.80	4.07±2.00	3.73±6.14	4.20±2.18	1.87±1.24
Exocanthion (left)	4.45±2.29	3.60±2.40	4.11±1.89	3.89±6.38	3.58±2.27	2.25±1.20
Palpebrale superius (right)	-	-	-	-	-	1.81±0.99
Palpebrale superius (left)	-	-	-	-	-	2.22±1.11
Palpebrale inferius (right)	-	-	-	-	-	1.80±0.99
Palpebrale inferius (left)	-	-	-	-	-	2.31±1.27
Glabella	6.35±3.32	-	-	-	-	2.70±1.58
Nasion	-	-	-	-	-	2.20±1.21
Pronasale	2.00±0.90	-	-	-	-	2.76±0.53
Subnasale	1.65±0.88	-	-	-	-	2.72±0.99
Subspinale	1.41±0.56	-	-	-	-	2.55±1.16
Alare (right)	4.20±1.63	4.10±2.20	3.62±1.91	3.43±3.74	4.98±2.63	2.71±1.01
Alare (left)	3.44±1.38	3.90±2.00	3.32±1.94	3.60±4.01	3.77±1.87	2.54±0.62
Labiale superius	1.51±0.71	3.50±2.50	4.19±2.34	3.09±3.06	2.94±1.35	2.92±1.51
Stomion	1.84±1.08	-	-	-	-	2.87±2.33
Labiale inferius	2.35±0.78	5.20±5.20	8.82±7.12	4.36±6.03	3.73±2.97	2.72±1.04
Christa philtra (right)	2.77±1.69	-	-	-	-	3.92±1.95
Christa philtra (left)	3.81±1.30	-	-	-	-	3.58±1.97
Cheilion (right)	1.93±0.93	4.90±3.60	7.52±4.57	3.76±4.05	3.94±2.96	2.65±1.30
Cheilion (left)	3.35±2.59	4.70±3.50	7.15±4.64	3.95±4.17	3.88±2.86	2.56±1.18
Sublabiale	4.34±3.22	-	-	-	-	3.29±1.41
Pogonion	3.50±2.94	-	-	-	-	3.70±1.65
Gnathion	4.85±3.10	-	-	-	-	4.56±1.22
Menton	-	-	-	-	-	2.76±0.93
Tragus (right)	-	-	-	-	-	1.82±0.36
Tragus (left)	-	-	-	-	-	1.72±0.49
Zygoin (right)	-	-	-	-	-	1.76±0.60
Zygoin (left)	-	-	-	-	-	1.24±0.52
Gonion (right)	-	-	-	-	-	4.03±1.42
Gonion (left)	-	-	-	-	-	3.49±0.72
Mean results	3.21±1.65	4.22±2.99	5.05±3.01	4.02±5.32	3.96±2.55	2.62±2.39

## Data Availability

The data presented in this study are available upon request from the corresponding author. The data are not publicly available due to the fact that the project creating this database is still in progress and has not received an open source license from the data owner.

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
