# Peer review of "An Automated Method of 3D Facial Soft Tissue Landmark Prediction Based on Object Detection and Deep Learning"

_diagnostics, 2023, doi:10.3390/diagnostics13111853_

Round 1
Reviewer 1 Report
1. This study proposes a prediction algorithm for facial soft tissue landmarks based directly on a 3D model of the human face, which can avoid various problems that occur when converting 3D models to 2D attribute maps.
2. The topic is relevant in the field.
3. The methodology is very good and correct.
4. The conclusions are consistent with the evidence.
5. The references are appropriate.
6. The obtained results demonstrated that the author’s novel method is more accurate and predicts more landmarks than other deep learning methods.
7. It is necessary to apply this method to a larger number of patients in order to obtain reliable results.
8. More research is definitely needed to establish this method in clinical practice.
Reviewer 2 Report
Paper well written and coherent with journal aims. However discussion should be improved to better highlight the advantages of this novel approach
Reviewer 3 Report
The manuscript proposes an algorithm for facial soft tissue landmakrs based directly on a 3D model of the human face.
The introduction is too long.
The aim is not clearly described and is chaotic.
IT is difficult to follow.
The error between the proposed method and manual labelling should be emphasised.
How did you calculate the error compared to the manual labelling(Table 3)?
The discussion is too short. It does not compare the data to others form the literature.
The conclusion is ambiguous and is not supported by the results.
English language should be revised
Round 2
Reviewer 3 Report
The paper has been improved, comments to reviewers answered.